# Large harvested energy with non-linear pyroelectric modules

Pierre Lheritier[1,4], Alvar Torelló[1,2,4 ✉], Tomoyasu Usui[3], Youri Nouchokgwe[1,2], Ashwath Aravindhan[1,2], Junning Li[1], Uros Prah[1], Veronika Kovacova[1], Olivier Bouton[1], Sakyo Hirose[3] & Emmanuel Defay[1 ✉]

Coming up with sustainable sources of electricity is one of the grand challenges of this century. The research field of materials for energy harvesting stems from this motivation, including thermoelectrics[1], photovoltaics[2] and thermophotovoltaics[3]. Pyroelectric materials, converting temperature periodic variations in electricity, have been considered as sensors[4] and energy harvesters[5–7], although we lack materials and devices able to harvest in the joule range. Here we develop a macroscopic thermal energy harvester made of 42 g of lead scandium tantalate in the form of multilayer capacitors that produces 11.2 J of electricity per thermodynamic cycle. Each pyroelectric module can generate up to 4.43 J cm⁻³ of electric energy density per cycle. We also show that two of these modules weighing 0.3 g are sufficient to sustainably supply an autonomous energy harvester embedding microcontrollers and temperature sensors. Finally, we show that for a 10 K temperature span these multilayer capacitors can reach 40% of Carnot efficiency. These performances stem from (1) a ferroelectric phase transition enabling large efficiency, (2) low leakage current preventing losses and (3) high breakdown voltage. These macroscopic, scalable and highly efficient pyroelectric energy harvesters enable the reconsideration of the production of electricity from heat.

Energy harvesting with pyroelectric materials requires the cycling of temperature with time, in contrast to the spatial temperature gradient needed for thermoelectric materials. This implies thermodynamic cycles, which are best described by entropy ($S$)–temperature ($T$) diagrams. Figure 1a shows the typical $S$–$T$ diagram of a non-linear pyroelectric (NLP) material exhibiting a phase transition representative of the ferroelectric–paraelectric electric-field-driven phase transition in lead scandium tantalate (PST)[8]. The area of the blue and green cycles drawn in the $S$–$T$ diagram corresponds to the converted electrical energy in an Olsen cycle (two isothermal and two isofield legs). Here we consider two cycles with the same electric field variation (field on and off) and with the same variation of temperature $\Delta T$, although not at the same initial temperature. The green cycle does not lie in the phase transition region and hence exhibits a much smaller area than the blue cycle, which is in the phase transition area. In $S$–$T$ diagrams, the larger the area, the larger the harvested energy. Therefore, phase transitions should enable more energy to be harvested. The need for large-area cycles in the case of NLPs is very similar to what is required for electrocalorics[9–12], for which multilayer capacitors (MLCs) of PST and poly(vinylidene fluoride)-based terpolymer have recently shown excellent solid-state cooling performances in reversed cycles[13–16]. Thus, we identified PST MLCs of interest for the purpose of thermal energy harvesting. These samples have been thoroughly described in the Methods and characterized as displayed in Supplementary Notes 1 (scanning electron microscopy), 2 (X-ray diffraction) and 3 (calorimetry).

## Energy density from thermodynamic cycles

The energy density $N_d$ that can be harvested in one cycle by a pyroelectric material is

$$N_d = \oint E\,\mathrm{d}D \tag{1}$$

where $E$ and $D$ are the electric and electric displacement fields, respectively. $N_d$ can be obtained indirectly from $DE$ loops (Fig. 1b) or directly by running thermodynamic cycles. The most useful ones were described by Olsen during the 1980s in its pioneering work on pyroelectric energy harvesting[17].

Figure 1b shows two unipolar $DE$ loops of a 1-mm-thick PST-MLC sample collected at 20 °C and 90 °C, respectively, both between zero and 155 kV cm⁻¹ (600 V). These two loops can be used to indirectly work out the energy harvested with the Olsen cycle displayed in Fig. 1a. Indeed, an Olsen cycle is constituted of two isofield legs (here with the zero field in leg DA and 155 kV cm⁻¹ in leg BC) and two isothermal legs (here 20 °C in leg AB and 90 °C in leg CD). The energy collected throughout the cycle corresponds to both the orange and blue areas (the integral of $E\,\mathrm{d}D$). The harvested energy $N_d$ is the difference between the input and output energies, that is, only the orange area in Fig. 1b. This particular Olsen cycle yields an energy density $N_d$ of 1.78 J cm⁻³. Stirling cycles are an alternative to Olsen cycles (Supplementary Note 7). Easier to implement because of the steps at constant charge (open circuit), the

[1]Materials Research and Technology Department, Luxembourg Institute of Science and Technology (LIST), Belvaux, Luxembourg. [2]University of Luxembourg, Esch-sur-Alzette, Luxembourg. [3]Murata Manufacturing Co., Ltd., Nagaokakyo, Japan. [4]These authors contributed equally: P. Lheritier, A. Torelló. ✉e-mail: alvar.torello@list.lu; emmanuel.defay@list.lu

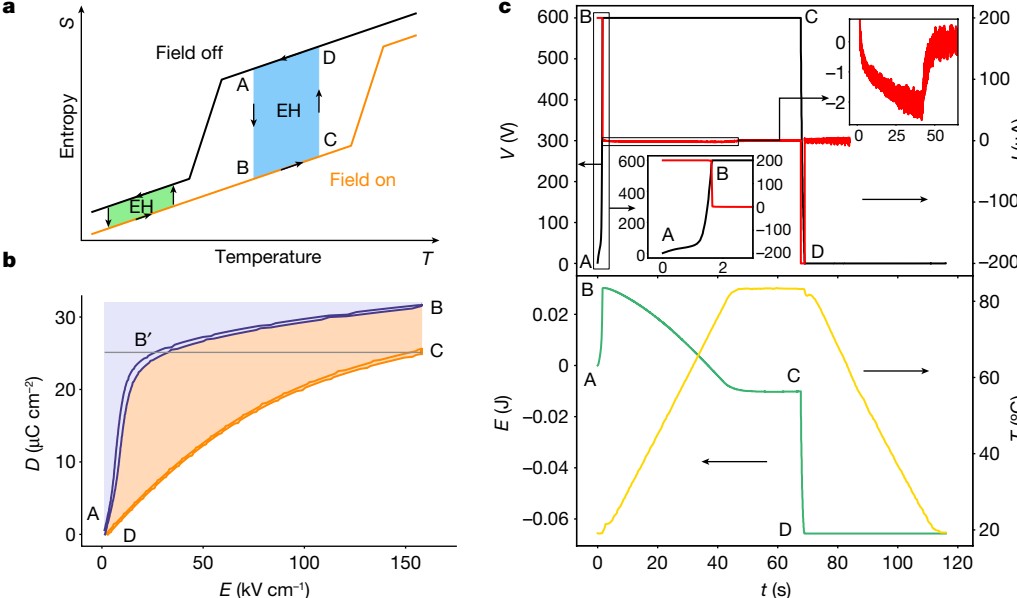

**Fig. 1 | Olsen cycle. a**, Sketch of an entropy (*S*)–temperature (*T*) diagram with electric field on and off applied to a NLP material exhibiting a phase transition. Two energy harvesting cycles are shown in two different temperature regions. The blue and green cycles occur in and out, respectively, of the phase transition, ending up in very different surface areas. **b**, Two unipolar *DE* loops of a 1-mm-thick PST MLC measured between 0 and 155 kV cm⁻¹ at 20 °C and 90 °C, respectively, and the corresponding Olsen cycle. The letters ABCD refer to the different states in the Olsen cycle. A–B: MLC charged up to 155 kV cm⁻¹ at 20 °C. B–C: MLC kept at 155 kV cm⁻¹ while the temperature increases up to 90 °C. C–D: MLC discharged at 90 °C. D–A: MLC cooled down to 20 °C at zero field. The

cyan area corresponds to the input electrical energy needed to run the cycle. The orange area is the energy harvested in one cycle. **c**, Top panel, voltage (black) and current (red) versus time monitored during the same Olsen cycle as in **b**. The two insets are enlargements of voltage and current at crucial moments in the cycle. In the bottom panel, the yellow and green curves are the corresponding temperature profile and energy, respectively, of the 1-mm-thick MLC. Energy is calculated from the current and voltage curves in the top panel. Negative energy corresponds to harvested energy. The capital letters in the four graphs correspond to the same steps as the Olsen cycle. Cycle AB'CD corresponds to a Stirling cycle (Supplementary Note 7).

energy density extracted from Fig. 1b (cycle AB'CD) reaches 1.25 J cm⁻³. This is only 70% of what can be harvested from an Olsen cycle, but it enables simple harvesting devices to be made.

In addition, we measured directly the energy harvested during an Olsen cycle by using a Linkam stage to control the temperature and a sourcemeter to impose voltage to a PST MLC (Methods). Figure 1c, top and the associated insets display the current (red) and the voltage (black) collected on the same 1-mm-thick PST MLC as that used for the *DE* loops experiencing the same Olsen cycle. Current and voltage enable the harvested energy to be worked out, for which the profile is shown in Fig. 1c, bottom (green) together with the temperature throughout the cycle (yellow). The letters ABCD stand for the same Olsen cycle throughout the whole of Fig. 1. The MLC charge occurs during the leg AB, conducted at low current (200 μA), so that the sourcemeter can properly monitor the charge. The result of this constant initial current is that the voltage profile (black curve) is not linear because of the non-linear electric displacement field *D* of PST (Fig. 1c, top inset). At the end of the charge, 30 mJ of electric energy is stored in the MLC (point B). The MLC is then heated up and a negative current appears (and therefore harvesting as well) while the voltage is kept at 600 V. After 40 s, this current cancels out as the temperature reaches a plateau at 90 °C, although during this isofield step the sample yielded back 35 mJ of electric work to the circuit (second inset Fig. 1c, top). Then the voltage is decreased across the MLC (leg CD), which further releases 60 mJ of electric work. The total output energy is then 95 mJ. The amount of harvested energy is the difference between the input and output energies, which yields 95 − 30 = 65 mJ. This corresponds to an energy density of 1.84 J cm⁻³, which is very close to $N_d$ extracted from the *DE* loops. The reproducibility of such Olsen cycles has been thoroughly checked (Supplementary Note 4). By pushing the voltage and temperature further, we reached 4.43 J cm⁻³ with an Olsen cycle in 0.5-mm-thick PST MLCs at 750 V (195 kV cm⁻¹) and a temperature

span of 175 °C (Supplementary Note 5). This is four times larger than the best value reported in the literature from direct Olsen cycles and obtained on Pb(Mg,Nb)O₃-PbTiO₃ (PMN-PT) thin films (1.06 J cm⁻³)[18] (see Supplementary Table 1 for more values from the literature). This performance has been reached owing to the very low leakage current of these MLCs (<10⁻⁷ A at 750 V and 180 °C, see details in Supplementary Note 6)—a crucial point mentioned by Smith et al.[19]—in contrast to the materials used in earlier studies[17,20].

The same conditions (600 V, 20–90 °C) were applied to a Stirling cycle (Supplementary Note 7). As expected from the *DE* loop results, the yield was 41.0 mJ. The most striking point of the Stirling cycles is their ability to amplify the initial voltage through the pyroelectric effect. We observed voltage amplification factors of up to 39 (from an initial voltage of 15 V to a final voltage reaching 590 V, see Supplementary Fig. 7.2).

## Macroscopic pyroelectric harvesters

Another notable feature of these MLCs is that they are macroscopic objects large enough to harvest energy in the joule range. Hence, we built a harvester prototype (HARV1) using 28 1-mm-thick PST MLCs following the same parallel plate design described in Torello et al.[14], structured in a 7 × 4 matrix, as schematically depicted in Fig. 2a. A dielectric fluid carrying heat in the harvester was displaced with a peristaltic pump between the two reservoirs in which the fluid was kept at constant temperature (Methods). Up to 3.1 J were harvested with the Olsen cycle described in Fig. 2a, with isothermal legs at 10 °C and 125 °C and isofield legs at 0 and 750 V (195 kV cm⁻¹). This corresponds to an energy density of 3.14 J cm⁻³. Measurements at different conditions were performed with this harvester (Fig. 2b). Note that 1.8 J were harvested with a temperature span of 80 °C and a voltage of 600 V (155 kV cm⁻¹). This fits very well with the 65 mJ mentioned earlier for one 1-mm-thick PST MLC under the same conditions (28 × 65 = 1,820 mJ).

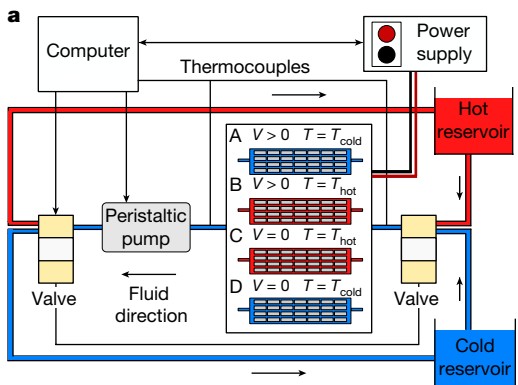

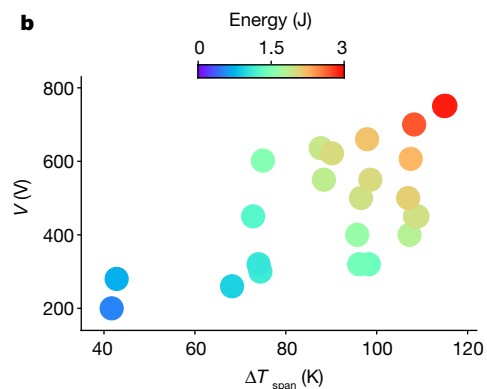

**Fig. 2 | Harvesting prototype. a**, Experimental set-up of harvesting prototype HARV1 based on 28 1-mm-thick PST MLCs (4 rows × 7 columns) running an Olsen cycle. For each of the four steps of the cycle, the temperature and voltage in the prototype are provided. A computer drives a peristaltic pump circulating a dielectric fluid between a cold and a hot reservoir, two valves and a power supply. The computer also collects voltage and current applied to the prototype from the power supply and the temperature of the harvester using thermocouples. **b**, Energy harvested (colours) by our 4 × 7 MLC prototype in different experiments versus temperature span (*x* axis) and voltage (*y* axis).

A larger version of the harvester (HARV2) with 60 1-mm-thick PST MLCs and 160 0.5-mm-thick PST MLCs (41.7 g of active pyroelectric material) yielded 11.2 J (Supplementary Note 8). In 1984, Olsen built an energy harvester based on 317 g of Sn-doped Pb(Zr,Ti)O$_3$ compounds able to generate 6.23 J of electricity around 150 °C (ref. [21]). This is the only other available value in the joule range for such a harvester. He obtained slightly more than half the value we reached with nearly seven times more mass. This means that the energy density of HARV2 is 13 times larger.

The cycle period in HARV1 is 57 s. This yields a power of 54 mW with the 4 rows × 7 columns set of 1-mm-thick MLCs. To go further, we built a third harvester (HARV3), with one single 0.5-mm-thick PST MLC and a similar set-up as for HARV1 and HARV2 (Supplementary Note 9). We measured a thermalization time of 12.5 s. This translates to a cycle period of 25 s (Supplementary Fig. 9). The harvested energy (47 mJ) infers an electric power of 1.95 mW per MLC, which in turn lets us envisage HARV2 generating 0.55 W (around 1.95 mW × 280 0.5-mm-thick PST MLCs). In addition, we simulated the heat exchange with finite-element modelling (COMSOL, Supplementary Note 10 and Supplementary Tables 2–4) matching the HARV1 experiment. Finite-element modelling enables us to foresee power values nearly one order of magnitude larger (430 mW) for the same amount of PST by thinning the MLCs down to 0.2 mm, using water as heat exchange fluid and reverting the matrix to 7 rows × 4 columns (960 mW when, in addition, the reservoir is next to the harvester, Supplementary Fig. 10b).

To prove the usefulness of such harvesters, a Stirling cycle was applied on an autonomous demonstrator that contains only two 0.5-mm-thick PST MLCs as the heat harvester, a high voltage switch, a low-voltage converter with storing capacitors, a d.c./d.c. converter, a low-consumption microcontroller, two thermocouples and a boost converter (Supplementary Note 11). This circuit needs an initial charge of its storage capacitors at 9 V and can then run autonomously as long as the temperature of the two MLCs varies between −5 °C and 85 °C, here in a cycle of 160 s (several cycles are displayed in Supplementary Note 11). It is notable that two MLCs weighing only 0.3 g can run this large system autonomously. Another interesting feature is the ability of the low-voltage converter to convert 400 V to 10–15 V with 79% efficiency (Supplementary Note 11 and Supplementary Fig. 11.3).

## PST MLC efficiency

We finally assessed how efficient these MLC modules are at converting thermal heat into electric energy. The efficiency's figure of merit $\eta$ is defined as the ratio of harvested electric energy density $N_d$ over input heat density $Q_{in}$ (Supplementary Note 12):

$$\eta = \frac{N_d}{Q_{in}} \tag{2}$$

It is also interesting to consider the Carnot efficiency

$$\eta_{Carnot} = \frac{\Delta T}{T_{hot}} \tag{3}$$

and the relative efficiency (or scaled efficiency) $\eta_r$ with respect to Carnot, which is

$$\eta_r = \frac{\eta}{\eta_{Carnot}} = \frac{N_d T_{hot}}{Q_{in} \Delta T} \tag{4}$$

Figure 3a,b show the efficiency $\eta$ and the scaled efficiency $\eta_r$, respectively, of the Olsen cycles as a function of the temperature span of one 0.5-mm-thick PST MLC. Both datasets are given for an electric field of 195 kV cm$^{-1}$. The efficiency $\eta$ reaches 1.43%, which corresponds to 18% of $\eta_r$. However, $\eta_r$ reaches values as large as 40% for a 10 K temperature span between 25 °C and 35 °C (the light blue curve in Fig. 3b). This is twice as large as the best known value in NLP materials, reported in PMN-PT thin films ($\eta_r = 19\%$) for a 10 K temperature span and 300 kV cm$^{-1}$ (ref. [18]). Temperature spans lower than 10 K were not considered because of the thermal hysteresis of PST MLCs, which is between 5 and 8 K wide. It is fundamental here to recognize the positive role played by the phase transition on efficiency. Indeed, the best values of $\eta$ and $\eta_r$ are nearly all obtained at the initial temperature $T_i = 25$ °C in Fig. 3a,b. This stems from the proximity of the phase transition when no field is applied, for which the Curie temperature $T_C$ is around 20 °C in these MLCs (Supplementary Note 13).

This last observation has two important consequences: (1) any efficient cycle should start at a temperature higher than $T_C$ to enable the occurrence of the field-induced phase transition (from paraelectric to ferroelectric) and (2) these materials are more efficient when operated close to $T_C$. Despite the large scaled efficiency displayed in our experiments, the limited temperature range prevents large absolute efficiencies to be reached because of the Carnot limitation ($\Delta T / T$). However, the outstanding efficiency displayed by these PST MLCs could demonstrate that Olsen was right when he mentioned that "an ideal 20-stage regenerative pyroelectric engine operating between 50 °C and 250 °C may be 30% efficient"[17]. To reach such values and validate the concept, it would be useful to use doped PST with different $T_C$ as studied by Shebanov and Borman. They showed that $T_C$ in PST can vary between 3 °C

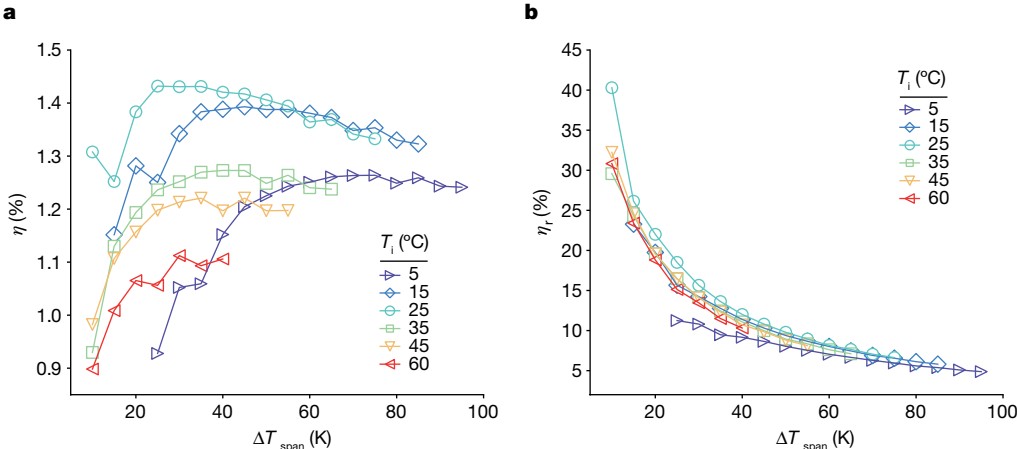

**a**

**b**

**Fig. 3 | Pyroelectric efficiency of PST MLCs. a,b,** The efficiency $\eta$ of the Olsen cycles (**a**) and the scaled efficiency $\eta_{\mathrm{r}} = \eta/\eta_{\mathrm{Carnot}}$ (**b**) of a 0.5-mm-thick PST MLC as a function of temperature span $\Delta T_{\mathrm{span}}$ for a maximum electric field of 195 kV cm$^{-1}$ and different initial temperatures $T_{\mathrm{i}}$.

(Sb doping) and 33 °C (Ti doping)[22]. We therefore envisage that the next generation of pyroelectric regenerators based on doped PST MLCs or other materials with a strong first-order phase transition could compete with the best energy harvesters.

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

# Article

## Methods

### PST-MLC fabrication

In this study, we worked on MLCs made of PST. These devices consist of a succession of Pt electrodes and PST in such a way as to obtain several capacitors all connected in parallel. PST has been chosen because it is an excellent EC material and therefore a potentially excellent NLP material. It exhibits a sharp ferroelectric–paraelectric first-order phase transition around 20 °C, which infers that its entropy varies similarly to what is represented in Fig. 1. Similar MLCs have already been described thoroughly for the purpose of EC devices[13,14]. In this study, we used $10.4 \times 7.2 \times 1\ mm^3$ and $10.4 \times 7.2 \times 0.5\ mm^3$ MLCs. The 1-mm- and 0.5-mm-thick MLCs are made of 19 and 9 inner layers, respectively, of 38.6-μm-thick PST. In both cases, the PST inner layers are intercalated between 2.05-μm-thick Pt electrodes. The design of these MLCs infers that 55% of PST is active, corresponding to the part in between the electrodes (Supplementary Note 1). The active electrode area is 48.7 mm² (Supplementary Table 5). MLCs of PST were prepared by solid-state reaction and tape casting methods. Details of the preparation processes were reported in a previous paper[14]. One of the differences of the PST MLCs from this previous paper[14] is the B-site ordering, which strongly influences EC performance in PST. B-site ordering of PST MLCs is 0.75 (Supplementary Note 2), obtained by sintering at 1,400 °C followed by a long annealing period of several hundreds of hours at 1,000 °C. More details of the PST MLCs are given in Supplementary Notes 1–3 and Supplementary Table 5.

### Experimental set-up of the Olsen cycles

The main concept of this study is based on Olsen cycles (Fig. 1). For such cycles, we need a cold and a hot heat reservoir and a power supply able to control and monitor the voltage and current in the different MLC modules. Two different configurations have been used for these direct cycles, namely (1) a Linkam module heating and cooling one single MLC associated with a Keithley 2410 energy supply and (2) three prototypes (HARV1, HARV2 and HARV3) based on several PST MLCs connected in parallel with the same energy supply. In the latter case, a dielectric fluid (silicone oil of viscosity 5 cP at 25 °C, purchased from Sigma Aldrich) was used to exchange heat between the two reservoirs (hot and cold) and the MLCs. The hot reservoir consisted of a glass vessel that contained the dielectric fluid and was placed on top of a hot plate. The cold reservoir consisted of a thermal bath of the fluid tube containing the dielectric fluid in a large plastic vessel filled with water and ice. Two three-way pinch valves (purchased from Bio-Chem Fluidics) were placed at each end of the harvester to properly switch the fluid flow from one reservoir to the other one (Fig. 2a). To ensure thermal equilibrium between the PST-MLC stack and the heat transfer fluid, the period of the cycle was extended until the inlet and outlet thermocouples (placed as closely to the PST-MLC stack as possible) read the same temperature. A Python script governed and synchronized all of the instrumentation (sourcemeter, pump, valves and thermocouples) so that proper Olsen cycles were run, that is, the hot fluid loop started circulating through the PST stack after the sourcemeter had charged them so that they were heated up at the desired applied voltage of a given Olsen cycle.

Alternatively, we confirmed these direct measurements of harvested energy with an indirect method. These indirect methods are based on electric displacement field (*D*)– electric field (*E*) loops collected at different temperatures and enabling an accurate estimation of how much energy can be harvested by calculating the area in between the two *DE* loops, as depicted in Fig. 1b. These *DE* loops were also collected with the Keithley sourcemeter.

### Harvester description

**HARV1.** Twenty-eight 1-mm-thick PST MLCs were assembled in a 4 row × 7 column parallel plate structure following the design described in ref. [14]. The fluid slit in between the PST-MLC rows was 0.75 mm. This was achieved by adding double-sided-tape stripes acting as fluid spacers at the edges of the PST MLCs. The PST MLCs were electrically connected in parallel with silver epoxy bridges that contacted the electrode terminals. After that, a wire was glued with silver epoxy to each side of the electrode terminal so that it could be connected to the power supply. Finally, the entire structure was inserted into a polyolefin hose. The latter was glued to the fluid tubes to ensure proper sealing. At the end, 0.25-mm-thick type K thermocouples were embedded at each end of the PST-MLC structure to monitor the temperature of the inlet and outlet of the fluid. To do so, the hose had to be first perforated. Once the thermocouple was embedded, the same glue as before was applied in between the hose and the thermocouple wire to restore the seal.

**HARV2.** Eight individual prototypes were built, four of them with 40 0.5-mm-thick PST MLCs each, distributed in 5 column × 8 row parallel plate structures, and the other four with 15 1-mm-thick PST MLCs each, distributed in 3 column × 5 row parallel plate structures. The total number of PST MLCs used was 220 (160 0.5-mm-thick and 60 1-mm-thick PST MLCs). We refer to these two subunits as HARV2_160 and HARV2_60. The fluid slit in HARV2_160 prototypes consisted of two stripes of 0.25-mm-thick double-sided tape and a 0.25-mm-thick wire in between them. For HARV2_60 prototypes, we repeated the same procedure but with 0.38-mm-thick wires instead. For the sake of symmetry, HARV2_160 and HARV2_60 had their own fluid circuit, pump, valve and cold side (Supplementary Note 8). The hot reservoir, a three-litre container (30 cm × 20 cm × 5 cm) on top of two hot plates with rotating magnets, was shared by the two HARV2 subunits. All eight individual prototypes were connected electrically in parallel. In the Olsen cycle that led to 11.2 J of harvested energy, subunits HARV2_160 and HARV2_60 were operated simultaneously.

**HARV3.** One single 0.5-mm-thick PST MLC was placed in a polyolefin hose, with double-sided tape stripes and wires at the sides to create spaces for the fluid to flow through. Because of its smaller size, the prototype was placed next to the valve that supplies fluid either from the hot fluid reservoir or the cold one, minimizing the cycle period.

### Olsen cycles

A constant electric field was imposed in the PST MLCs by applying a constant voltage throughout the heating leg. As a result, a negative pyroelectric current was generated and energy was harvested. After the PST MLCs were heated up, the field was removed (*V* = 0), and the energy stored in them was brought back to the sourcemeter, which corresponds to another contribution of the harvested energy. Finally, at the applied voltage *V* = 0, the PST MLCs were cooled down to their initial temperature so that a cycle could start again. In this step, no energy was harvested. We ran Olsen cycles with a Keithley 2410 sourcemeter by charging the PST MLCs at the voltage source and setting the current compliance to the appropriate value so that enough points in the charging step were gathered to enable a reliable calculation of the energy.

### Stirling cycles

In Stirling cycles, PST MLCs were charged in voltage source mode at an initial electric field value (initial voltage $V_i > 0$), a desired compliance current so that the charging step takes around 1 s (and enough points are gathered for a reliable calculation of the energy) and cold temperature. Before the PST MLC was heated up, the electric circuit was opened by imposing current compliance *I* = 0 mA (the lowest value of current compliance that our sourcemeter could take was 10 nA). As a result, charges were kept in the PST MLCs and voltage increased while the sample was heated up. In the leg BC, no energy was harvested because *I* = 0 mA. After reaching the hot temperature and the voltage in the PST MLCs having been amplified (in some cases it was by more than 30 times, see Supplementary Fig. 7.2), the PST MLCs were discharged

($V = 0$), and the electrical energy stored in them was brought back to the sourcemeter at the same current compliance that they had been charged with initially. Because of the voltage amplification, the energy stored at the hot temperature was higher than the energy supplied at the beginning of the cycle. Thus, energy was harvested by converting heat into electrical energy.

## Calculation of energy harvested and power

We monitored the voltage and current applied to the PST MLCs with a Keithley 2410 sourcemeter. The corresponding energy was calculated by integrating the product of voltage and current read by the Keithley sourcemeter over time, $E = \int_0^\tau I_{\mathrm{meas}}(t)\, V_{\mathrm{meas}}(t)$, where $\tau$ is the period of the cycle. In our energy curves, positive values of energy mean energy we have to supply to the PST MLCs, and negative values mean energy we extract from them, hence harvested energy. The associated power of the given harvested cycle was deduced by dividing the harvested energy by the period of the entire cycle $\tau$.

## Data availability

All data is available in the main text or the Supplementary Information. Correspondence and requests for materials should be addressed to A.T. or E.D. Source data are provided with this paper.

**Acknowledgements** We thank N. Furusawa, Y. Inoue and K. Honda for their assistance in fabricating the MLCs. P.L., A.T., Y.N., A.A., J.L., U.P., V.K., O.B. and E.D. acknowledge the Fonds National de la Recherche (FNR) of Luxembourg for supporting this work through the projects CAMELHEAT C17/MS/11703691/Defay, MASSENA PRIDE/15/10935404/Defay-Siebentritt, THERMODIMAT C20/MS/14718071/Defay and BRIDGES2021/MS/16282302/CEC0HA/Defay.

**Author contributions** E.D. suggested the entire experimental study. S.H. and T.U. prepared the PST-MLC samples. P.L. ran the experiments that led to Fig. 1. A.T. prepared the experimental set-up and run the experiments that led to Fig. 2. P.L. gathered the *DE* loops that led to Fig. 3. Y.N. ran the scanning electron microscopy and differential scanning calorimetry (DSC) experiments that lead to Supplementary Notes 1–3. P.L. studied the reproducibility of Olsen cycles and ran the experiments that led to Supplementary Note 4. J.L. ran the experiments that led to Supplementary Note 5 with A.T. J.L. ran the leakage study that led to Supplementary Note 6 and EC characterization in Supplementary Note 12. A.T. ran the Stirling cycles in Supplementary Note 7 and the heat exchange simulations that led to Supplementary Note 10. A.T. and U.P. designed and built the HARV2 experiment. A.T. and U.P. ran the HARV2 experiment with J.L., A.A., Y.N. and V.K. E.D. and O.B. designed the autonomous device system with V.K. O.B. developed the electric circuitry and microcontroller script that governs the autonomous system. V.K. and A.A. ran the experiments of the autonomous system that led to Supplementary Note 11 with A.T. E.D., P.L., A.T. and V.K interpreted the key findings. P.L. designed the python script that ran Olsen and Stirling characterizations and HARV1, HARV2 and HARV3 devices with A.T. E.D., P.L. and A.T. elaborated the efficiency equations in Supplementary Note 12. E.D. gathered the data and constructed Supplementary Table 1. A.T. built Supplementary Tables 2–5. P.L. and A.T. curated the data. A.T. prepared the figures. E.D. wrote the manuscript with P.L. and A.T. Y.N. and V.K. contributed to the final version of the manuscript. Y.N., A.A. and J.L. contributed to the final version of the Supplementary Information. E.D. obtained the funding and supervised the project.

**Competing interests** The authors declare no competing interests.

**Additional information**
**Correspondence and requests for materials** should be addressed to Alvar Torelló or Emmanuel Defay.
