## [Peer Review File · Nature]

Manuscript Title: Large harvested energy with non-linear pyroelectric modules

Reviewer Comments & Author Rebuttals

Reviewer Reports on the Initial Version:

Referees' comments:

Referee #1 (Remarks to the Author):

The paper is a very nice and well written report on the use of multi-layer devices for pyroelectric harvesting based on an Olsen type cycle. The experiments are undertaken in a logical and rigorous manner with a very elegant setup (Fig2a). The attractive performance is related to the low loss of the material/devices and operation around the phase transition which good energy density compared to the literature.

A multi-layer structure is used - does the use of multiple layers enhance the power compared to a monolithic element of the same volume - this is not clear and could be more clear. For example, as a piezoelectric energy harvester, a multilayer increases the charge, but reduces the open-circuit voltage compared to a monolithic configuration resulting in the same energy. Details of the origin and manufacture of the multi-layer devices could be more clear.

This is an excellent piece of rigorous work and deserves to be published in a high-quality journal - however, the actual novelty of the work is less clear for a journal such as Nature; for example Olsen developed the cycles in the 1980s and operation of the Olsen cycle around the phase transition to improve harvested energy is also known and reported e.g. Phase transformation based pyroelectric waste heat energy harvesting with improved practicality, Hwan Ryul Jo and Christopher S Lynch 2016 Smart Mater. Struct. 25 035009. In its current form, the paper is a very high-quality implementation of the Olsen approach - moving it to the closer practical application of the technology.

Referee #2 (Remarks to the Author):

In this study P. Lheritier and co-workers investigated the pyroelectric energy harvesting performance of well known PST based ferroelectric MLC and the arrayed module by using widely known Olsen cycle. The performance of the MLC and the module is noticeably good and their analysis does not have critical defects. However there is no significantly impressive fundamental findings or potential for practical applications in this work. Authors mentioned that phase transition temperature, leakage, pyroelectric/electrocaloric effect of the material is crucial factors, but there are no data on these. Reviewer believes that the work will be of interest to a subset of the scientific community, but

it may have difficulties to be deployed to practical applications because of inevitably applied high field to utilize large harvested energy.

1. To extract the energy from Olsen cycle, it is believed that high bias electric field switching with temperature changes are definitely necessary. How can we supply such high field to the devices in real situation? To operate DC source for such high voltage switching device, the energy for DC power source driving would be more than harvested energy from the module. In addition, we have to extract the electric current from the device under high bias voltage, how can we do that? It should be clarified. Though authors showed current generation data from the MLC and module, please show the charging of battery/capacitor or driving of practical devices such as electric motor, light bulb, electronic devices, etc.

2. For harvesting performance test, it is necessary to show the result of multiple cycle test for the durability and repeatability.

3. Authors showed the schematic drawings of the device and test set-up. In addition to these, please provide photos of module and test set-up. It would be more helpful for readers. What was the base substrate material for the module? Does it not affect the performance of the MLC module?

4. How did authors calculate power of the device? To calculate the proper power value, authors have to provide testing conditions such as electric load for voltage and current measurements and both should be measured simultaneously. Multiplication of max voltage (open circuit or applied voltage) and max current (short circuit current) is not proper way. 54 mW from 28 MLC (9g) is not exceptionally high if it is compared to other harvesting mechanisms. It would be better to show driving of small electric devices with authors' harvesting module to confirm the generated power.

5. There are lack of detailed information about the material used in this work although authors mentioned the references. Please add summary table/figure of materials properties (dielectric properties, leakage behavior, phase transition curve, etc) of PST used in this study in SI. And provide full composition of the material.

6. Minor issues:

in Fig. 2 b and Table 1, it should be 'delta Tscan'

Terminologies 'PE' and 'DE' are inconsistently used in the manuscript.

Method, Active electrode area was approximately 49 mm²/layer -> 49 mm²/layer

Referee #3 (Remarks to the Author):

Major revisions :

In this work, the authors introduced a method to realize Olsen cycle during the ferroelectric-paraelectric phase transition of PST. The phase transition temperature of PST is 17 °C and the mentioned device can be applied around room temperature. The energy conversion efficiency is obviously improved twice higher compared with previous reported works. This is a meaningful and novel work that deserves publication in Nature. However, there are still some issues needed to be

noted and it can be published after major revision. The comments are as follows:

1. Other definition and expression of energy conversion efficiency and other methods to improve pyroelectric performance were reported in literature. (Adv. Electron. Mater. 2018, 1800413; Adv. Mater. 2019, 1902831; Nano Energy 55 (2019) 534-540) The authors should cite and describe these related contents in the manuscript.
2. The physical properties of ferroelectric materials have an abrupt change at ferroelectric-paraelectric phase transition temperature. Why does the best performance of this work obtained at the temperature 8 °C higher than phase transition temperature? The temperature-dependence curves of related physical properties of PST-MLC around Curie temperature should be measured.
3. Did the authors detect the temperature of the PST-MLC directly? Was the temperature of PST-MLC consistent with that of dielectric fluid? How to overcome the temperature measurement hysteresis of thermocouples?
4. How to realize the synchronous change of temperature and voltage? (Fig. 2a) the authors should give a brief introduction in this manuscript.
5. The S-T or D-E curves correspond to the energy harvesting cycles mention in the manuscript (such as Line 134) should be provided in Supplementary Materials.

Author Rebuttals to Initial Comments:

Preliminary comment from the authors:

The authors would like to thank all reviewers for their overall positive and constructive feedback. We have made efforts to improve the quality of our work and pushed further our initial findings. In this reviewed version, we have added three new accomplishments: 1) the successfully upgraded version of our harvester prototype, harvesting 11.2 J with 42 g of active material – because of that, we have changed the title of the manuscript to “A 10 Joule-pyroelectric harvester”-, 2) a larger harvested energy density of 4.43 J cm^{-3} (four times larger than the best value reported in the literature) by enlarging the temperature range; and 3) the continuous and autonomous operation of a sensor, powered exclusively with the energy harvested by 0.3 g of pyroelectric material, demonstrating the development of real applications with this technology.

Referee #1 (Remarks to the Author):

The paper is a very nice and well written report on the use of multi-layer devices for pyroelectric harvesting based on an Olsen type cycle. The experiments are undertaken in a logical and rigorous manner with a very elegant setup (Fig2a). The attractive performance is related to the low loss of the material/devices and operation around the phase transition which good energy density compared to the literature.

Answer: The authors thank reviewer 1 for such nice and positive feedback.

A multi-layer structure is used - does the use of multiple layers enhance the power compared to a monolithic element of the same volume - this is not clear and could be clearer. For example, as a piezoelectric energy harvester, a multilayer increases the charge, but reduces the open-circuit voltage compared to a monolithic configuration resulting in the same energy.

Answer: Indeed, the multi-layer configuration increases the charge, and for the same applied voltage, we can reach much higher electric fields because of the much thinner active layer. Recently, we have shown that the maximum field we could apply to a bulk PST was 40 kV cm^{-1} , whereas up to 290 kV cm^{-1} were shown possible to be applied in very similar PST-MLCs (Nair et al, Nature 2019). As a result, the energy stored in the MLCs is much more.

Changes in the manuscript – We now describe better the performances of our MLCs but adding four dedicated notes in supplementary (Note 1. MLC structure; Note 2. X-Ray diffraction; Note 3. Calorimetric measurements, Note 6. Leakage current in PST-MLCs). We also compare the results obtained in the literature on monolithic structures and our MLCs in Supplementary Table 1. The improvement in energy density in our case is at least 5 fold.

Details of the origin and manufacture of the multi-layer devices could be clearer.

Answer: We appreciate this fair comment from Referee #1. We now have added descriptions and more details of our PST-MLCs in supplementary material. This includes XRD analysis, DE loops, entropy change with field and SEM images detailing the inner structure and dimensions.

Changes: SM notes 1 MLC structure, SM note 2 XRD analysis and SM note 12 efficiency of Olsen cycles.

This is an excellent piece of rigorous work and deserves to be published in a high-quality journal –

Answer: The authors would like to thank reviewer 1 for such a positive and constructive feedback.

however, the actual novelty of the work is less clear for a journal such as Nature; for example, Olsen developed the cycles in the 1980s.

Answer - The actual novelty of our work is a complete revisitation of the topic with bespoke nonlinear pyroelectric modules with a material exhibiting 1) a transition close to room temperature, 2) extremely low leakage current and 3) much larger applied electric fields. This is a game changer in the reported values compared to the closest state of the art. We obtained an energy density 13 times larger than what Olsen reported at the time, thanks to these unique MLCs. Besides, something we have improved since the first version of this paper is the fabrication of a bigger energy harvester able to collect 11.2 J of electrical energy, twice as much as what Olsen reported with 7 times less active material. Moreover, we built a totally new sustainable device by using a Stirling cycle.

Changes – This comment induced most of the changes implemented in the paper, which are numerous, and all reported in blue in the text of the main text. Moreover we added notes in supplementary to report on the Stirling cycles (Note 7), on the bigger energy harvester giving 11.2 J (Note 8) and on the autonomous energy harvester (Note 11).

Also, operation of the Olsen cycle around the phase transition to improve harvested energy is also known and reported e.g. Phase transformation based pyroelectric waste heat energy harvesting with improved practicality, Hwan Ryul Jo and Christopher S Lynch 2016 Smart Mater. Struct. 25 035009.

Answer: The authors appreciate the comment from the reviewer. In the paper the reviewer cited (Hwan Ryul Jo and Christopher S Lynch 2016 Smart Mater. Struct. 25 035009), the authors claim that “A net energy density of 0.27 J cm⁻³ per cycle was obtained from the ferroelectric material using the modified cycle with a temperature change between 25°C and 180°C.” This is more than 16 times less than what we obtained in our work. In addition, there is no practical application displayed in their publication. Our modules have been made on purpose for this very application, and PST is an excellent material for that. We therefore believe that the uniqueness of our pyroelectric modules make all the difference in this context.

In its current form, the paper is a very high-quality implementation of the Olsen approach - moving it to the closer practical application of the technology.

Answer: We thank reviewer 1 for such positive feedback about our work. Indeed, we have now successfully proved the operation of an application with only 0.3 g of active pyroelectric materials.

Changes: In SM note 11: Autonomous energy harvester.

Referee #2 (Remarks to the Author):

In this study P. Lheritier and co-workers investigated the pyroelectric energy harvesting performance of well-known PST based ferroelectric MLC and the arrayed module by using widely known Olsen cycle. The performance of the MLC and the module is noticeably good, and their analysis does not have critical defects.

Answer: The authors would like to thank Referee #2 for the positive feedback.

However, there is no significantly impressive fundamental findings or potential for practical applications in this work.

Answer: Indeed, we did not display a practical application with our previous work. We have work hard and included now in the manuscript a self-sufficient device able to power an autonomous sensor with only 0.3 g of active pyroelectric material. This includes a complete description of the electronic circuitry as well as pictures of the device, schematics and diagrams and experimental data recorded.

Besides, we believe that the actual novelty of our work is a complete revisit of the topic with bespoke nonlinear pyroelectric modules with a material exhibiting 1) a transition close to room temperature, 2) extremely low leakage current and 3) much larger applied electric fields. This is a game changer in the reported values. Besides, we have developed an upgraded version of our first prototype, harvesting 11.2 J, twice as much as what Olsen reported with 7 times less active material.

Changes – This comment, together with one from Reviewer 1, induced most of the changes implemented in the paper, which are numerous, and all reported in blue in the text of the main text. Moreover we added notes in supplementary to report on the Stirling cycles (Note 7), on the bigger energy harvester giving 11.2 J (Note 8) and on the autonomous energy harvester (Note 11).

Authors mentioned that phase transition temperature, leakage, pyroelectric/electrocaloric effect of the material is crucial factors, but there are no data on these.

Answer: The authors appreciate this fair comment from Referee #2. We have now included more material and information in supplementary, as calorimetry measurements showing the phase transition, a study of the leakage current of the PST-MLCs and the electrocaloric characterisation. In fact, we have included the EC effect contribution in our efficiency definition. In addition, we have measured ourselves the intrinsic properties of the 0.5 mm thick PST-MLCs, such as volumetric density, specific heat, latent heat and active volume.

Changes: in SM note 4 “Calorimetry measurements”, in SM note 6 “Leakage current”, and in SM note 12 “Efficiency of Olsen cycles in PST-MLC”

Reviewer believes that the work will be of interest to a subset of the scientific community, but it may have difficulties to be deployed to practical applications because of inevitably applied high field to utilize large, harvested energy.

Answer: The authors appreciate this fair comment from Referee #2. We specifically developed one application (autonomous sensor) and used another thermodynamic cycle (Stirling) that is much less demanding in terms of high voltage. We lose around 30% in energy density, but it strongly simplifies the overall system, as described in this new version.

Changes (in manuscript): “Stirling cycle is an alternative to Olsen cycle. Easier to implement thanks to steps at constant charge (open circuit), the energy density extracted from Fig. 1b (cycle AB’CD) reaches 1.25 J cm^{-3} . This is only 70% of what can be harvested from an Olsen cycle, but it enables making simple autonomous devices, as disclosed later in this article.” (...) “In order to prove the usefulness of such harvesters, a Stirling cycle was applied on an autonomous demonstrator that contains only two 0.5mm-thick MLCs as heat harvester, a high voltage switch, a low voltage converter with storing capacitors, a DC/DC converter, a low consumption microcontroller, two thermocouples and a boost converter (see SM note 11 for details). This circuit needs an initial charging at 9 V of its storage capacitors and can then run autonomously as long as the temperature of the two MLCs varies between -5°C and 85°C in a cycle of 160 s (several cycles displayed in SM). It is remarkable that two MLCs weighing only 0.3 g can run this large system autonomously. Another remarkable feature is the ability of the low voltage converter, made of an inductance and storage capacitors, to convert voltage from large (400 V) to low (10-15V) voltage with 79 % efficiency.”

Moreover, a complete description of this autonomous energy harvester is provided in Supplementary Note 11.

1. To extract the energy from Olsen cycle, it is believed that high bias electric field switching with temperature changes are necessary. How can we supply such high field to the devices in real situation? To operate DC source for such high voltage switching device, the energy for DC power source driving would be more than harvested energy from the module.

Answer: The authors appreciate the comment from Referee #2. It is difficult indeed, though possible because our modules can survive large fields. But Stirling cycles are much less demanding because high field last for much shorter time, and the charging leg of the cycle is at lower fields / voltages. All in all, there is no need to keep voltage on all the time, only the charge. As already mentioned, we implemented such self-charging device and included all the details in supplementary.

Changes: SM note 7 “Stirling cycles” and SM note 11 “Autonomous energy harvester”.

In addition, we have to extract the electric current from the device under high bias voltage, how can we do that? It should be clarified. Though authors showed current generation data from the MLC and module, please show the charging of battery/capacitor or driving of practical devices such as electric motor, light bulb, electronic devices, etc.

Answer: The authors appreciate the comment from Referee #2 that triggered a tremendous amount of work in our team in order to realize this low voltage transduction and improve its efficiency. As already commented in previous answers, the new autonomous energy harvester we fabricated can harvest 79% of the energy created in the pyro modules at high voltage (up to 600 V) and lower it at a practical voltage (less than 20V) thanks to a proper electronic circuitry we developed by transmitting the electrical energy harvested (at high voltage) to storage capacitors (low voltage) through an inductor. It works very well. All the details are gathered in supplementary.

Changes: SM note 11 “Autonomous energy harvester”.

2. For harvesting performance test, it is necessary to show the result of multiple cycle test for the durability and repeatability.

Answer: The authors appreciate this fair comment from Referee #2. Indeed, this is shown now in supplementary note 4, where 20 Olsen cycles have been run consecutively. The average energy and its statistical error are 21.80 ± 0.03 mJ, which shows its durability and repeatability.

Changes: SM note 4 “Reproducibility of Olsen cycles”.

3. Authors showed the schematic drawings of the device and test set-up. In addition to these, please provide photos of module and test set-up. It would be more helpful for readers. What was the base substrate material for the module? Does it not affect the performance of the MLC module?

Answer: The authors appreciate the constructive comment from Referee #2. Now we have included photos of the experimental set-up in supplementary, showing the fluid circuitry, hot and cold reservoirs, the piezoelectric modules and the fluidic pumps. We have also included more details of the PST-MLCs and SEM images displaying their cross-section and inner structure. The MLC modules do not have any substrate.

Changes: SM note 1 “MLC structure”, SM note 8 “HARV2 experimental set-up and design”

4. How did authors calculate power of the device? To calculate the proper power value, authors have to provide testing conditions such as electric load for voltage and current measurements and both should be measured simultaneously. Multiplication of max voltage (open circuit or applied voltage) and max current (short circuit current) is not proper way.

Answer: The authors appreciate the comment from Referee #2. In all these experiments, we collected simultaneously voltage and current, meaning that we actually have access to the real power at all time. We have now a better description of the set ups, notably the note 8 and 10, dedicated to the new energy harvesters developed in this new version. Moreover, a new thorough description of the harvesters with measurements conditions is provided in the methods section.

Changes: “We have monitored the voltage and current applied to the PST-MLCs with a Keithley 2410 sourcemeter. The corresponding energy has been calculated from integrating the product of voltage and current read by the Keithley over time $E = \int_0^t I_{meas}(t)V_{meas}(t)$. In our energy curves, positive values of energy mean energy we have to supply to the PST-MLCs, and negative values mean energy we extract from them, hence, energy harvested. The associated power of the given harvested cycle has been deduced by dividing the energy harvested by the period of the entire cycle”.

54 mW from 28 MLC (9g) is not exceptionally high if it is compared to other harvesting mechanisms. It would be better to show driving of small electric devices with authors' harvesting module to confirm the generated power.

Answer - The 54 mW of the 28 1 mm-MLC device (labelled now as HARV1) were calculated by diving the overall energy harvested (3,1 J, obtained from numerically integrating voltage and current data gathered through the entire cycle over time) by the total period cycle (57 s). In addition, we have improved our discussion by adding one extra experiment (labelled HARV3) to address better this important question. It consists of 1 single 0.5 mm PST-MLC, which shortens the thermalisation time respect to the temperature of the fluid. The total period of HARV3 is 24 s and the energy is 47 mJ. This yields an overall power of 1.96 mW per MLC. We also have added a new section in methods where we explicitly explain how energy and power are calculated from our raw data.

Changes: SM note 9 “HARV1 heat exchange numerical modelling” and SM note 10 “HARV3”

5. There are lack of detailed information about the material used in this work although authors mentioned the references. Please add summary table/figure of materials properties (dielectric properties, leakage behaviour, phase transition curve, etc) of PST used in this study in SI. And provide full composition of the material.

Answer: The authors thank Referee #2 for these fair points. We now have included much more details in Supplementary on the leakage behaviour, phase transition curves with calorimetry measurements and inner structure and composition of PST-MLCs.

Changes: SM note 1 “MLC structure”, SM note 2 “XRD Analysis”, SM note 3 “Calorimetry measurements” and SM note 6 “Leakage current”.

6. Minor issues:

in Fig. 2 b and Table 1, it should be ‘delta Tscan’

Terminologies ‘PE’ and ‘DE’ are inconsistently used in the manuscript.

Methods, Active electrode area was approximately 49 mm /layer -> 49 mm²/layer

Answer: The authors thank Referee #2 for spotting these inconsistencies. We have changed and fixed them. Now we use in all the text electric displacement field D terminology instead of polarization P . Thanks to our SEM study, we have now measured more precisely the active area of our PST-MLC, being it 48.7 mm². We have also corrected ΔT_{span} where indicated by the referee.

Changes (In methods): “...Active electrode area is 48.7 mm²/layer...”

Referee #3 (Remarks to the Author):

Major revisions:

In this work, the authors introduced a method to realize Olsen cycle during the ferroelectric-paraelectric phase transition of PST. The phase transition temperature of PST is 17 °C and the mentioned device can be applied around room temperature. The energy conversion efficiency is obviously improved twice higher compared with previous reported works. This is a meaningful and novel work that deserves publication in Nature. However, there are still some issues needed to be noted and it can be published after major revision.

Answer: The authors would like to thank Referee #2 for the positive feedback and for considering our work is publishable in Nature.

The comments are as follows:

1. Other definition and expression of energy conversion efficiency and other methods to improve pyroelectric performance were reported in literature. (Adv. Electron. Mater. 2018, 1800413; Adv. Mater. 2019, 1902831; Nano Energy 55 (2019) 534-540) The authors should cite and describe these related contents in the manuscript.

Answer: Indeed, there are other definitions and applications related to pyroelectric materials. We introduced two of these references in the abstract, where we think they fit best.

2. The physical properties of ferroelectric materials have an abrupt change at ferroelectric-paraelectric phase transition temperature. Why is the best performance of this work obtained at the temperature 8 °C higher than phase transition temperature? The temperature-dependence curves of related physical properties of PST-MLC around Curie temperature should be measured.

Answer: The authors appreciate the comment from Referee #3. Indeed, the transition temperature shifts towards higher temperatures with the applied voltage. We have provided in supplementary DE loops as well as entropy curves from calorimetry measurements showing this behaviour around the Curie temperature.

Changes: SM note 3 “Calorimetry measurements” and SM note 12 “Efficiency of Olsen cycles”, which include the DE loops.

3. Did the authors detect the temperature of the PST-MLC directly? Was the temperature of PST-MLC consistent with that of dielectric fluid? How to overcome the temperature measurement hysteresis of thermocouples?

Answer: The authors thank Referee #3 for this comment. The temperature of the PST-MLCs in the Linkam stage was recorded by IR camera. The temperature of the experimental set-up in Figure 2a was recorded with thermocouples placed very close to the stack of 28 PST-MLC,

measuring the temperature of the fluid at the very entry and at the very end of the PST-MLC stack. To ensure that the PST-MLCs reached the temperature of the fluid, we waited for both thermocouples to read the same temperature. When inlet and outlet thermocouples display the same temperature, it shows that thermal equilibrium has been reached in the PST-MLC stack and that the fluid does not alter its temperature when flowing through it. We have clarified that better in the methods section.

Changes (In the methods section): “To ensure thermal equilibrium between the PST-MLC stack and the heat transfer fluid, the period of the cycle is extended until the inlet and outlet thermocouples (placed as closely to the PST-MLC stack as possible) read the same temperature.”

4. How to realize the synchronous change of temperature and voltage? (Fig. 2a) the authors should give a brief introduction in this manuscript.

Answer: The authors appreciate the comment of Referee #3 and having the opportunity to explain better the experimental set-up. The experiment described in Figure 2a is run with a Python script which governs all instrumentation used. This includes the sourcemeter that supply charges and voltage to the PST-MLC stack, the peristaltic pump that moves the fluid and the pinched valves that switch the fluid circuitry from a hot to a cold loop and vice versa. When the instruction to charge the PST-MLC stack is sent to the power supply, another one is sent to the valves, that activate the fluid hot loop so that hot fluid starts circulating through the PST-MLC stack, heating them up. We have explained that better in the methods section.

Changes (In method section): “A Python script governs and synchronises all instrumentation (Sourcemeter, pump, valves, thermocouples, etc.) so that proper Olsen cycles are run, i.e., the hot fluid loop starts circulating through the PST-stack after the sourcemeter has charged it. Hence, the PST-stack is heated up while the desired applied voltage is being applied (isofield heating, leg BC in Olsen cycles described). The other cycle steps are run the same way.”

5. The S-T or D-E curves correspond to the energy harvesting cycles mention in the manuscript (such as Line 134) should be provided in Supplementary Materials.

Answer: The authors would like to thank Referee #3 for this fair comment. We have included in supplementary calorimetry data displaying the S-T curves at 0-field and at certain electric fields applied. The entropy displayed is given in arbitrary without units because our PST-MLC is too massive (0.3 g) for our calorimetry (Differential Scanning Calorimetry) to provide trustful quantitative values (maximum mass of the sample should be 20 mg). Pieces of PST-MLCs that are less than 20 mg are possible to measure, but only at 0 applied field. Despite this issue, the qualitative behaviour of the curves we show is correct. We also show the full spectrum of the DE-loops in supplementary.

Changes: SM note 3 “Calorimetry measurements” and DE-loops in SM note 12 “Efficiency of Olsen cycles in PST-MLCs”.

Reviewer Reports on the First Revision:

Referees' comments:

Referee #1 (Remarks to the Author):

I have read the reply to the referees and the paper is improved. The paper has high-quality results, but is clearly building on existing work and done in a very well thought out and careful manner. Based on the reply and referee, operation at phase transition for thermal harvesting has been undertaken and Olsen cycles have been studied before and therefore I still do not see any major novelty for a journal such as Nature.

Referee #2 (Remarks to the Author):

The manuscript presents high pyroelectric energy harvesting performance (over 10 J) by assembling 60 lead scandium tantalate (PST) multilayer capacitors (MLC). For practical applications authors used modified Stirling cycles instead of Olsen cycles. The quality of revised manuscript is improved, specifically, the autonomous harvesting system with author's MLC module section is impressive. However the reviewer thinks that the novelty of this manuscript is not fulfil the standard of Nature. The main concern is that there is no noticeable breakthrough technique or science in authors work. The materials are already known as cited by authors, and it is a kind of combination the known material, structure, and harvesting fundamentals. Especially authors claimed that high performance of energy harvesting (over 10 J) was obtained, but it is from parallel combination of 60 MLCs. If the number of MLC were more, the energy would be more than 100 J easily. Thus the reviewer believe that the 10 J from the module does not have strong impact for the community. It would better to emphasize energy density and efficiency rather than total energy from the assembled module.

Below are minor comments from reviewer.

- In abstract and conclusion; authors addressed that it had potential for harvesting device from the solar energy, but it is not demonstrated in the manuscript.
- One sentence summary, 40g -> 42 g
- Page 2. Abbreviation of TE (Thermoelectric?) should be defined
- It is rather difficult to follow the manuscript. Authors mentioned many times that voltage and current of the samples, it should be clearly mentioned whether it is input or output voltage/current. For example, Fig. S.1, 'the voltage in the sample increases to 600V at C', is it input or output voltage? No title and unit in y-axis (RHS)
- Authors mentioned that the thickness of the layer is 1 and 0.5 mm, but Fig. 1.2 shows 0.42 mm. It should be consistent. The PST is B-sited ordered. How authors can control the B-site ordering in PST?
- Page 6, 'Stirling cycles is their ability to amplify initial voltage in open circuit conditions. We observed voltage amplification factors up to 39' Authors applied the input voltage of 15 V to the MLC then is increased to 590V? What is the mechanism of this amplification?
- In HARV testing, MLC having thinner thickness (0.5 mm) has shorter cyclic time because of smaller

volume?

- Page 37, Description of the autonomous pyroelectric energy harvesting system, The min operating voltage of LTC 3588 might not be 4V. It is operational at 3 V.
- Please provide simple dielectric vs temp plot to show phase transition of the PST.
- If authors can provide comparison table for energy density (J/cm³ or J/g) with other harvesting technology (solar, vibration, thermoelectric, etc), it would be great to understand the superiority of energy density from authors work.

Referee #3 (Remarks to the Author):

I recommend to accept this manuscript. The revisions and responses are reasonable.

Author Rebuttals to First Revision:

Remaining comments from Referee #2 (Remarks to the Author):

The manuscript presents high pyroelectric energy harvesting performance (over 10 Joule) by assembling 60 lead scandium tantalate (PST) multilayer capacitors (MLC). For practical applications authors used modified Stirling cycles instead of Olsen cycles. The quality of revised manuscript is improved, specifically, the autonomous harvesting system with author's MLC module section is impressive. However, the reviewer thinks that the novelty of this manuscript is not fulfil the standard of Nature. The main concern is that there is no noticeable breakthrough technique or science in author's work. The materials are already known as cited by authors, and it is a kind of combination the known material, structure, and harvesting fundamentals. Especially authors claimed that high performance of energy harvesting (over 10 J) was obtained, but it is from parallel combination of 60 MLCs. If the number of MLC were more, the energy would be more than 100 J easily. Thus, the reviewer believes that the 10 J from the module does not have strong impact for the community. It would better to emphasize energy density and efficiency rather than total energy from the assembled module.

Answer: We thank reviewer 2 for this comment. Indeed, we forgot to mention that these modules are scalable, which opens the opportunity to build large prototypes. We now mention it in the initial summary.

Changes: "These new macroscopic, scalable, and highly efficient pyroelectric energy harvesters enable reconsidering the production of electricity from heat."

Below are minor comments from the reviewer.

- In abstract and conclusion; authors addressed that it had potential for harvesting device from the solar energy, but it is not demonstrated in the manuscript.

Answer: We have removed these comments, although we believe our findings clearly reveal the potential of these pyroelectric modules.

- One sentence summary, 40g -> 42 g

Answer: We have removed the one sentence summary.

- Page 2. Abbreviation of TE (Thermoelectric?) should be defined.

Answer: Yes, indeed. We have removed this acronym.

- It is rather difficult to follow the manuscript.

Answer: We have added headings to help navigate the readership through the article.

- Authors mentioned many times that voltage and current of the samples, it should be clearly mentioned whether it is input or output voltage/current.

Answer: The confusion comes certainly from the Stirling cycles, in which the final voltage is not the same as the initial voltage. Therefore, we use now initial and final labels rather than input and output throughout the paper. Accordingly, we have modified labels in figures S7.1, S7.2 and S11.4.

For example, Fig. S.1, 'the voltage in the sample increases to 600V at C', is it input or output voltage?

Answer: There is neither input nor output because this is a single capacitor. This is voltage that varies with time, in the cycle.

No title and unit in y-axis (RHS)

The authors do not understand this comment. We have carefully checked all figures from the main text and supplementary and they all have title and units in x and y axis.

- Authors mentioned that the thickness of the layer is 1 and 0.5 mm, but Fig. 1.2 shows 0.42 mm. It should be consistent.

Answer: We appreciate this fair point from reviewer 2. We added "(called 0.5 mm in the main paper, for sake of simplicity)" in the caption of Fig. 1.2 to make it consistent.

- The PST is B-sited ordered. How authors can control the B-site ordering in PST?

Answer: We appreciate this comment from reviewer 2. This is performed by changing the annealing conditions. We added the annealing conditions in "Methods: PST - MLCs fabrication", though they were reported elsewhere, as cited in the initial version "B-site ordering of PST-MLCs used in this study was 0.75, obtained by sintering them at 1400°C followed by a long annealing of several hundreds of hours at 1000°C."

- Page 6, 'Stirling cycles is their ability to amplify initial voltage in open circuit conditions. We observed voltage amplification factors up to 39' Authors applied the input voltage of 15 V to the MLC then is increased to 590V? What is the mechanism of this amplification?

Answer: We appreciate these questions from reviewer 2. The first question about input voltage probably stems from the same confusion about initial and final voltages. In Stirling cycles, the initial voltage (15 V for example) is amplified to 590 V thanks to the pyroelectric effect. Since polarisation decreases with temperature, voltage must increase if charge is maintained constant (open circuit conditions). We have rephrased our sentence in the manuscript underlying that this mechanism is the pyroelectric effect.

Changes: "The most striking point of Stirling cycles is their ability to amplify the initial voltage through the pyroelectric effect. We observed voltage amplification factors up to 39 (from an initial voltage of 15 V to a final voltage reaching 590 V, Fig. S7.2)."

- In HARV testing, MLC having thinner thickness (0.5 mm) has shorter cyclic time because of smaller volume?

Answer: Yes, indeed. The thinner thickness (and smaller volume then of MLCs) increases the heat exchange surface of the samples, shortening the time of reaching thermal equilibrium (Thermalization). Hence, we can shorten the cycle period accordingly.

- Page 37, Description of the autonomous pyroelectric energy harvesting system. The min operating voltage of LTC 3588 might not be 4V. It is operational at 3 V.

Answer: In effect, the input voltage of an LTC3588 may vary from 2.7 V to 20 V, depending on the module's configuration. In order to ensure output voltage of at least 3.6 V, we had to configure the LTC modules so that the minimum input voltage was 4 V.

- Please provide simple dielectric vs temp plot to show phase transition of the PST.

Answer: We have provided this in supplementary note 13.

- If authors can provide comparison table for energy density (J/cm³ or J/g) with other harvesting technology (solar, vibration, thermoelectric, etc), it would be great to understand the superiority of energy density from authors' work.

Answer: We thoroughly investigated the referee's comment and realized that we can indeed compare energy density with vibrating energy harvesters because they are based on equivalent charging/discharging cycles, which gives the opportunity to consider how much energy can be harvested in one cycle in $J\ cm^{-3}$. However, the notion of energy density does not really make sense in the case of photovoltaic or thermoelectric devices because they are both working on steady state effects, which call for power density in $W\ cm^{-2}$. Therefore, we added a small paragraph after supplementary Table 1 to compare with vibrating energy harvesters. It reads as follows.

Changes (in Supplementary Table 1): "The one-shot energy density that we obtained on our pyroelectric modules in Olsen cycles ($4.43\ J\ cm^{-3}$) can be directly compared with typical energy densities obtained with vibration energy harvesters. In Godard et al.^[6], we compared piezoelectric materials in the context of vibration energy harvesters. The best one is based on lead zirconate titanate (PZT) and can harvest at maximum $80\ mW\ cm^{-3}$ at 120 Hz. This stands for $6.5\ 10^{-4}\ J\ cm^{-3}$ per stroke, which means four orders of magnitude less than in our pyroelectric PST MLCs.